# Patients with Very High Risk of Cardiovascular Adverse Events during Carfilzomib Therapy: Prevention and Management of Events in a Single Center Experience

**DOI:** 10.3390/cancers15041149

**Published:** 2023-02-10

**Authors:** Giulia Mingrone, Anna Astarita, Anna Colomba, Cinzia Catarinella, Marco Cesareo, Lorenzo Airale, Arianna Paladino, Dario Leone, Fabrizio Vallelonga, Sara Bringhen, Francesca Gay, Franco Veglio, Alberto Milan

**Affiliations:** 1Hypertension Unit, Department of Medical Sciences, Division of Internal Medicine, AO “Città della Salute e della Scienza” University Hospital, 10126 Turin, Italy; 2SSD Clinical Trial in Oncoematologia e Mieloma Multiplo, Division of Hematology, AOU Città della Salute e della Scienza di Torino, 10126 Turin, Italy; 3Division of Hematology, Department of Molecular Biotechnology and Health Sciences, University of Torino, 10124 Turin, Italy

**Keywords:** cardiovascular toxicity, multiple myeloma, carfilzomib, cardiovascular adverse event, arterial hypertension, echocardiography, cardiotoxicity management

## Abstract

**Simple Summary:**

The use of carfilzomib in multiple myeloma is burdened by cardiovascular toxicity. Currently, the risk stratification of cardiovascular adverse events appears well established, while the prognosis of patients classified as being at high risk or experiencing adverse events is uncertain. We aimed to provide practical insight on the prevention and management of cardiovascular adverse events during carfilzomib therapy, taking advantage of the experience of our specialized center in managing patients with increased cardiovascular risk. After baseline evaluation, we shared a diagnostic workup, eventually including advanced cardiac imaging testing to identify inducible ischemia in patients with a high-cardiovascular-risk profile. We aimed for timely identification and treatment of underlying conditions and prevention of major cardiovascular events. With the same purpose, we suggest a management protocol for the main cardiovascular adverse events based on the presenting symptoms.

**Abstract:**

Carfilzomib (CFZ) improves the prognosis of multiple myeloma (MM) patients but has shown cardiovascular toxicity. The risk stratification of cardiovascular adverse events (CVAEs) now seems well established, while little is known about the course and management of patients with a high-cardiovascular-risk profile or experiencing CVAEs during therapy. Therefore, we aimed to describe our experience in decision making to support health professionals in selecting the best management strategies to prevent and treat CVAEs. A total of 194 patients with indication to CFZ underwent baseline evaluation of CVAEs risk and were prospectively followed. We propose a novel approach, which includes advanced cardiac imaging testing for patients at high baseline CV risk to rule out clinical conditions that could contraindicate starting CFZ. After baseline evaluation, 19 (9.8%) patients were found at high risk of CVAEs: 13 (6.7%) patients underwent advanced cardiac testing and 3 (1.5%) could not receive CFZ due to CV contraindications. A total of 178 (91.7%) patients started CFZ: 82 (46%) experienced arterial-hypertension-related events and 37 (20.8%) major CVAEs; 19 (10.7%) patients had to discontinue or modify the CFZ dosing regimen. Along with baseline risk stratification, subsequent cardiovascular clinical events and diagnostic follow-up both provided critical data to help identify conditions that could contraindicate the anticancer therapy.

## 1. Introduction

Carfilzomib (CFZ) is a second-generation proteasome inhibitor (PI) utilized in multiple myeloma (MM), both in the relapsed/refractory setting as well as in the induction/consolidation of newly diagnosed MM [1]. The PIs block the proteasome functions by stopping proteolysis, leading to the accumulation of intracellular proteins and apoptosis [2,3]. Myeloma cells are especially sensitive to PI action due to their high production and secretion of proteins. CFZ improves progression-free survival of MM patients and has a more favorable safety profile than other PIs but has shown major cardiovascular (CV) toxicity [4,5,6,7,8,9,10,11,12]. The mechanism of CV damage has not been entirely elucidated. It is probably due to the accumulation of unfolded, damaged and undegraded proteins that induce apoptosis and to the endothelial dysfunction caused by the misregulation of nitric oxide (NO) homeostasis [3,13,14]. Arterial hypertension, new onset or worsening heart failure, ischemic heart disease, chest pain, arrhythmias and dyspnea are the most common CVAEs reported in the literature [7,15,16]. A recent study from our group showed that 44.9% of MM patients treated with CFZ experienced one or more CVAEs, with 30.9% grade 3 or greater in severity (Common Terminology Criteria for Adverse Events, CTCAE, ≥3), and 14.7% experienced major CVAEs (defined as acute coronary syndrome, typical chest pain, post-infusion dyspnea, syncope, arrythmia or sudden cardiac death) [17]. The great observed incidence of CVAEs is probably due to an overlap of risk factors: the PI cardiotoxicity, the increased CV risk of an aging population and the multiple MM-related comorbidities. Moreover, CFZ seems to lead to early left ventricle (LV) function impairment, demonstrated by global longitudinal strain (GLS) impairment and diastolic dysfunction on echocardiography [18,19]. Although most CV complications are reversible, in some cases, they may lead to life-threatening complications and cause withdrawal or remodulation of anticancer therapy. Despite the well-established risk of CVAEs during CFZ therapy, no validated protocol for CV event management is currently available. Therefore, the treatment and prevention of CVAEs are not uniformly managed in patients receiving CFZ. In a previous study, we proposed a risk score for predicting the incidence of CVAEs during CFZ [17]. Developing a scoring system that identifies high-risk patients allows the clinician to apply different strategies to prevent CVAEs. On this basis, we shared a novel diagnostic approach with an advanced cardiac imaging test to identify inducible ischemia for patients with high baseline CV risk or suspect of coronary stenosis to allow early identification and treatment of underlying cardiac conditions, prevent major CVAEs and potentially reduce therapy withdrawal. Through our direct experience in terms of decision making regarding patients at high CV risk or after major CVAEs, we aim to provide real-world practical evidence to support health professionals in selecting the best management strategies to prevent and/or treat the CVAEs of MM patients during CFZ treatments, considering the impact on clinical outcome and hematologic therapy.

## 2. Materials and Methods

### 2.1. Study Design

From January 2015 to September 2022, patients > 18 years old with a diagnosis of MM followed by the Myeloma Unit (“Città della Salute e della Scienza”, Turin, Italy) with clinical indication to CFZ therapy underwent baseline evaluation before starting CFZ treatment in order to estimate the risk of CVAEs. Patients for whom the indication was confirmed and who started receiving CFZ underwent periodic follow-up during chemotherapy. The study protocol was approved by the interagency ethic committee of “A.O.U. Città della Salute e della Scienza” hospital of Turin, Italy (protocol number 0038655), and each patient signed a written consent form. Patients affected by hematological disease other than multiple myeloma or by light chain cardiac amyloidosis (assessed by end-organ biopsy or cardiac magnetic resonance imaging) and those refusing enrolment were excluded. All patients underwent a comprehensive CV evaluation at our EchoLab (Hypertension Unit, University of Turin, Italy) before starting CFZ infusions. The CV evaluation consisted of clinical anamnestic assessment, office blood pressure (BP) measurement, pulse wave velocity (PWV) measurement, ambulatory blood pressure monitoring (ABPM), electrocardiogram (ECG) and transthoracic echocardiography (TTE). Given these hemodynamic and CV evaluations, we applied the CFZ CVAEs risk score equation [17] and divided our population into three risk classes: low risk (≤33%), intermediate risk (>33% and ≤66%) and high risk (>66%). The baseline evaluation was completed with an advanced cardiac imaging test, such as coronary computed tomography angiography (CCTA), myocardial single photon emission computed tomography (SPECT) and cardiac magnetic resonance (CMR) [20,21], in patients considered at very high CV risk in the presence of a high calculated CVAEs risk score and/or relevant risk factors for CVAEs during therapy (such as uncontrolled office BP, left ventricular hypertrophy, impaired GLS and increased PWV) [17] or if there was a high suspicion of coronary artery disease.

All patients were re-evaluated after 6 months of CFZ therapy, but if any CVAE occurred, the follow-up visit was anticipated, and the subsequent follow-up was modified on an individual basis.

For all patients, the clinical follow-up continued to the end of therapy with CV revaluation every 6 months in the case of high/intermediate CV risk and every 12 months in the case of low CV risk.

### 2.2. Cardiovascular Assessments

A detailed description of the study methodology has been reported in our previous works [17,18] (see also Appendix B).

Office BP measurements were performed with an automatic sphygmomanometer (Omron, M10-IT model). Office BP was considered controlled if the average BP was <140/90 mmHg, and in the case of uncontrolled BP values at baseline, antihypertensive treatment was started or optimized according to the latest guidelines [22]. ABPM was performed with a 24 h recording using a validated device (Takeda TM2430; A&D Company Ltd., Tokyo, Japan). Arterial stiffness estimation was performed through carotid-femoral pulse wave velocity (PWV) measurement (Sphygmocor system Atcor Medical, Sydney, Australia) [23]. A value ≥ 9 m/s was considered the cut-off for an increased risk of CVAEs according to our previous studies [17,24]. Two-dimensional TTE and speckle-tracking echocardiography (STE) analysis were performed following current guidelines at baseline and during follow-up [25,26]. Standard 2D images were acquired with an iE33, Affinity 50 or EPIQ7C ultrasound machine (Philips Medical System, Andover, MA, USA) equipped with a sector probe (S5-1 transducer). Left ventricular hypertrophy (LVH) was defined in the presence of LV mass ≥ 115 g/m^2^ and ≥95 g/m^2^ in men and women, respectively. LV diastolic function was defined according to the current cut-offs [26]. Speckle-tracking analysis was performed with commercially available software (Automated Cardiac Motion Quantification, QLAB Cardiac Analysis, Philips, Andover, MA, USA) [27,28]. A GLS value ≤ −20% was considered normal. Cardiotoxicity was defined as a new decline in LVEF below 40%, a new LVEF reduction by ≥10 percentage points to an LVEF of 40–49% or a new relative decline in GLS by 15% from baseline, according to the latest guidelines on cardiotoxicity [29].

### 2.3. Cardiovascular Adverse Events Definition

The incidence of CVAEs was detected both at the CV evaluations and through periodic review of patient medical records from the hematology unit. CVAEs were assessed and graded according to the Common Terminology Criteria for Adverse Events (CTCAE) version 5.0 [30]. We divided the CVAEs into arterial-hypertension-related events and non-hypertension-related (or “major”) events. (Definitions of CVAEs used are available in Appendix C).

### 2.4. Statistical Analysis

Statistical analysis was performed by using Jamovi program (the Jamovi Project 2022 Version 2.3.18.0, computer software). Quantitative variables were expressed as mean values and standard deviation or median values and interquartile ranges according to their distribution (assessed by Shapiro–Wilk test). Qualitative variables were expressed as absolute values and percentages. Unpaired Student’s *t*-test or Mann–Whitney test were performed for comparisons of baseline characteristics for quantitative variables, as appropriate. Kaplan–Meier curves with the cumulative hazard function were utilized to represent the incidence of hypertension-related and major CVAEs during follow-up. *p* < 0.05 was assumed as the level of statistical significance for all analyses.

## 3. Results

Out of 227 patients evaluated at our center, 194 met the inclusion criteria and were included in analyses (see the flowchart of the study population in Appendix A). The baseline characteristics of our population are listed in Table 1. The mean age was 67.1 ± 8.4 years, and there was a similar proportion of male and female patients. Tobacco use, arterial hypertension and obesity were the most common CV risk factors. Nearly half of the patients at the time of the first evaluation were on antihypertensive therapy. The median time from MM diagnosis was 4.1 [1.6–6.9] years, and 90.2% of patients had been treated with multiple oncologic therapies before CFZ (see Table 1).

### 3.1. Baseline Cardiovascular Risk Evaluation

According to baseline office BP and/or ABPM measurements, 83 patients (42.8%) were diagnosed with arterial hypertension, requiring the introduction of new antihypertensive drugs or modulation of the previous therapy.

Subclinical CV organ damage was evaluated: LVH was present in 34 (17.5%) patients, GLS impairment in 40 (20.6%) patients and increased arterial stiffness in 58 (29.9%) subjects.

The CV risk score was available in 124 patients (63.9% of the population): 19 (15.3%) patients were assigned a high CV risk, 75 (60.5%) intermediate risk and 30 (24.2%) low risk of CV adverse events during CFZ.

A total of 13 patients (6.7%) were identified at very high baseline risk of CVAEs during CFZ because of the presence of a high CVAEs risk score and/or relevant risk factors for CVAEs during therapy (such as uncontrolled office BP, LVH, impaired GLS and increased PWV). These patients underwent advanced cardiac imaging before starting treatment with CFZ (see the hemodynamic and echocardiographic parameters at baseline in Table 2).

Nine patients performed CCTA as a first-tier diagnostic test, four of which had critical coronary lesions. Subsequently, one patient was referred directly to coronary angiography (refused by the patient) and three patients underwent myocardial SPECT. In two cases, SPECT was positive for inducible ischemia and subsequent coronary angiographies were performed.

Two patients underwent myocardial SPECT as their first ischemia test and tested positive for inducible ischemia; further testing was not performed due to clinical worsening.

Similarly, a single patient underwent ECG stress testing first, showing ischemic ECG changes; further investigations were avoided due to disease progression.

Only one patient underwent CMR as a first-tier test for cardiac disease; results were normal.

Out of 13 patients who underwent advanced testing, 10 started CFZ having excluded major cardiac contraindications and 3 could not start CFZ therapy (see Appendix A).

### 3.2. CVAES Incidence and Management

A total of 178 (91.7%) patients began CFZ therapy and 48.7% of them experienced major or hypertension-related CVAEs during a median follow-up of 9.1 [4.35–18.48] months. Moreover, 12.9% of patients experienced both hypertensive-related and major CVAEs.

#### 3.2.1. Hypertension-Related CVAEs

A total of 46% of patients had hypertension-related CVAEs 3.5 [0.93–6.8] months after starting CFZ on average (see Figure 1).

Specifically, 38.8% of patients experienced new onset or worsening arterial hypertension, 23.6% had arterial hypertension just before the CFZ infusion (contraindicating the infusion in 10.7%), 12.9% had arterial hypertension after CFZ infusion and 3.4% had uncontrolled hypertension with related symptoms; no hypertensive emergencies were reported (see Table 3).

A total of 42.1% of the hypertensive adverse events were classified as CTCAE grade ≥ 3.

In 53 (64.6%) patients with hypertension-related events, antihypertensive therapy was introduced or increased; in one patient, after a severe hypotensive event, therapy was reduced. After the CV office re-evaluation, in 11 (13.4%) patients with hypertension-related CVAEs, a 24 h ABPM was required, and in 5 cases, the BP values were over the normal range. One patient had uncontrolled BP post CFZ infusion with concomitant left arm paraesthesia; therefore, ECG and myocardial enzymes were required, testing negative for signs of ischemia. Four patients (4.9%) with hypertensive CVAES had to modulate the CFZ therapy because of hypertensive events; of these, two patients discontinued CFZ and two patients reduced the dose (see Appendix A).

#### 3.2.2. Major CVAES

A total of 20.8% of patients experienced major CVAEs after 3.63 [0.96–7.03] months from starting CFZ (see Figure 1): 3.9% had dyspnoea, 6.7% arrythmia (seven atrial fibrillation, one atrial bigeminy, one nonsustained ventricular tachycardia, two ventricular bigeminy/trigeminy and one atrioventricular block), 3.4% a severe hypotensive event, 6.2% heart failure, 4.5% typical chest pain, 4.5% acute coronary syndrome (one ST elevation and seven non-ST elevation myocardial infarction), 0.6% syncope and 0.6% sudden cardiac death (see Table 3).

A total of 25.5% of the major cardiac events were classified as CTCAE grade ≥3.

Further testing was deemed necessary in 28 (75.5%) patients who experienced a major CVAE. In eight (23.5%) cases, markers of myocardial injury (troponin and creatin kinase MB) were dosed either because of chest pain or heart failure symptoms; in all these cases, myocardial ischemia could be ruled out. One patient underwent an exercise stress test after experiencing chest pain with hypotension, showing negative results. In 13 (35.1%) patients, a CCTA was performed, with 3 patients showing significant coronary artery stenosis who underwent subsequent coronary angiography. Myocardial SPECT was performed in two patients, one of whom tested positive for inducible ischemia.

Out of all patients with major CVAEs, six (16.2%) underwent coronary angiography (three cases after CCTA, one after myocardial SPECT and two direct coronary angiographies) and four patients had lesions that were deemed critical and treated with revascularization.

Five patients underwent CT angiography of the chest because of dyspnea and/or chest pain, with three negative results, one case of pulmonary embolism (PE) and one case of interstitial pneumonia.

Overall, 11 (29.7%) patients of those experiencing major CVAEs had to discontinue treatment with CFZ due to CV toxicity, and 4 (10.8%) continued CFZ on a reduced dosing regimen.

#### 3.2.3. Cardiotoxicity on Echocardiography

During follow-up, all patients have been monitored for potential cardiac organ damage with periodic TT echocardiograms. Echocardiographic signs of cardiac toxicity were reported in 16 (9%) patients: 3 (18.7%) showed significant LVEF reduction by ≥10 percentage points to an LVEF of 40–49% and 14 (87.5%) a decline in GLS by >15% from baseline. Six (37.5%) patients underwent CCTA, which tested negative for significant coronary stenosis in all cases; one patient performed myocardial SPECT with negative results. Among these patients, five (31.3%) discontinued treatment with CFZ.

## 4. Discussion

CFZ has shown a higher frequency of CVAEs than others PI; therefore, it is essential to allow the continuation of CFZ while trying to prevent the occurrence of CVAEs and give the appropriate treatment to MM patients [31]. No validated protocol for the management of CV events in this setting is available. For this reason, we shared a novel approach, which includes advanced cardiac imaging testing to improve risk stratification and reduce the incidence of major CVAEs associated with CFZ therapy. Moreover, we described our experience in terms of symptom-based management to facilitate early identification and treatment of the main major CVAEs.

### 4.1. Prevention of CVAEs

A total of 48.7% of our study population experienced hypertensive or major CVAEs, therefore, control of CV risk factors is an essential step in order to reduce the occurrence of CVAEs and allow patients to begin or continue treatment with CFZ.

The incidence of adverse events in MM patients during CFZ was increased by age, global frailty and the presence of multiple CV risk factors. Moreover, the observed incidence of arterial hypertension in our sample was greater than previously reported in the literature [10], probably because of our attention as a specialized center (Hypertension Unit).

Before starting a potentially cardiotoxic therapy, a baseline CV evaluation is crucial to estimate the individual CV risk and to optimize the cardioprotective therapy [12,32,33,34]. In a previous work, we proposed the only prospectively validated approach to the prevention of CVAEs; such a strategy relies on calculating a CV risk score based on systolic office BP, BPV at ABPM, arterial stiffness (estimated by PWV), LVH and left ventricular GLS assessed at baseline [17]. To further improve CV risk stratification and potentially reduce the incidence of major CVAEs, we shared a novel approach with advanced cardiac imaging testing for patients with very high CV risk or with clinical suspicion of coronary disease. The most appropriate diagnostic test should be decided according to the clinical indication, center-specific experience and availability. Out of 13 very-high-risk patients who underwent advanced cardiac imaging testing, 3 were deemed to have major cardiac contraindications to CFZ therapy. All patient candidates for CFZ had an incurable hematologic disease; therefore, the risk of clinical deterioration due to the withdrawal of anticancer therapies and the feasibility of alternative treatments should be carefully evaluated with the hematologists. For this purpose, the shared strategy could assist clinicians in detecting patients with real cardiovascular contraindications to CFZ among those with very high CV risk at baseline, thus minimizing withdrawal of treatment. In the presence of critical coronary stenosis, every single case must be managed in agreement with the cardiologist and/or cardiotoxicity specialist and the beginning/continuation of CFZ therapy should be discussed with the hematologist, following a multidisciplinary approach.

### 4.2. Management of CVAEs

Due to the lack of a standardized management approach to CVAEs during oncologic therapy, we try to promote a simple symptom-based protocol for CVAEs according to our real-life experience (see Figure 2).

All patients with clinical indication to CFZ therapy should undergo a baseline evaluation of their CV risk profile in order to estimate the likelihood of CVAEs during therapy and must be clinically followed to detect any symptoms or signs of CVAEs. All patients were re-evaluated after 6 months of CFZ therapy, but the follow-up visit was brought forward if any CVAE occurred, and the subsequent follow-up was modified on an individual basis.

Out of 178 (91.7%) patients who began CFZ, 95 (48.7%) experienced major or hypertension-related CVAEs during a median clinical follow-up of 9 months. Despite the significant incidence of CVAEs in our sample, with adequate follow-up and management, only a few patients had to discontinue CFZ therapy: 4 patients for hypertension-related events and 15 for major CVAEs. The most frequently reported symptoms that may suggest major CVAEs were chest pain, dyspnea and severe hypotensive events.

Chest pain occurred quite commonly; patients with clinical suspicion of coronary artery disease should undergo further cardiac testing according to their clinical presentation and center-specific availability (CCTA, ECG stress test, myocardial SPECT, CMR or direct coronary angiography). In our population, 13 CCTA and 2 myocardial SPECT were performed; 6 patients underwent coronary angiography (3 cases after CCTA, 1 after myocardial SPECT and 2 direct coronary angiographies), and in 4 cases, lesions were critical and treated with revascularization. A total of 13 patients had to discontinue or modify the dosing regimen. Cases that test positive for cardiac ischemia should be discussed with the cardiologist or with a clinician experienced in cardiotoxicity to assess the need for further testing or procedures. Once the cardiac workup is completed, patients should be re-assessed with the hematologist in order to either confirm the appropriateness of the current regimen, modify it or discontinue CFZ.

Shortness of breath and fatigue are commonly reported by MM patients. During the CV evaluation, the clinician should focus attention on the clinical features of dyspnea. Five chest contrast-enhanced CTs were performed for dyspnea and chest pain in our sample: three were negative, one confirmed PE and one led to a diagnosis of interstitial pneumonia. Shortness of breath during exertion should suggest ischemic heart disease and/or heart failure, but in some patients, dyspnea with a sudden onset or related to exertion is due to arrhythmias such as atrial fibrillation, which can be detected during CV assessment with 12-lead ECG or ambulatory ECG monitoring. Sudden-onset dyspnea may also suggest PE, and a chest contrast-enhanced CT should be ordered. In presence of accompanying symptoms such as fever or cough, pneumonia should be suspected, and a chest X-ray or chest CT should be performed.

During CFZ therapy, signs and symptoms of heart failure such as dyspnea, orthopnea and lower extremity edema should be carefully investigated: in our population, 11 (6.2%) of all patients experienced heart failure during therapy. If clinically indicated, a complete CV re-evaluation should be performed, investigating the possibility of an underlying coronary disease. In all cases, medical therapy for heart failure has to be optimized following the current guidelines [35].

The presence of an unexpected hypotensive event (systolic BP <90 mmHg and diastolic BP < 60 mmHg) or syncope in a previous normotensive or hypertensive patient could be of concern. Six episodes of severe hypotension occurred in our sample: one was caused by PE, one by critical coronary stenosis subsequently treated with revascularization and one occurred a few days before sudden death. Therefore, the clinical relevance of unexpected hypotensive events should not be underestimated, and the clinical presentation together with the results of a CV-focused clinical assessment should guide the diagnostic strategy.

Our study has several limitations. Firstly, our approach with advanced cardiac imaging testing in very-high-risk patients is a novel strategy not yet supported by evidence and it was applied only to a subsample of our population because it was recently introduced. The absence of a control group with similar age and risk factors greatly limits the scientific evidence of the proposed approach with advanced cardiac imaging testing for patients with defined CV risk factors. On the other hand, the potential toxicity of CFZ and related risk of therapy withdrawal encourage a thorough patient selection. Further studies, possibly including a control group, are needed to determine the long-term consequences on the reduction in the incidence of major CVAEs. Furthermore, coronary patency was tested in few patients; most of them underwent CCTA instead of myocardial SPECT or perfusion CMR due to our center-specific availability. Consequently, a comparison between different imaging techniques was not possible. Moreover, because of the lack of sufficient data on laboratory biomarkers (NT-proBNP and cardiac troponins) and lipid profiles, it was not possible to test their prognostic significance.

## 5. Conclusions

Baseline CV evaluation is crucial to estimate the individual CV risk and to optimize cardioprotective therapy. In order to improve risk assessment, we proposed a novel strategy based on advanced cardiac imaging testing for selected patients at high CV risk in an effort to identify and treat underlying cardiac conditions, thus preventing major CVAEs. With the same purpose, we suggested a management protocol for main CVAEs based on the presenting symptoms. Thus, the prevention protocol for high-CV-risk patients and the management of patients presenting with symptoms probably due to cardiotoxicity should be standardized.

## Figures and Tables

**Figure 1 cancers-15-01149-f001:**
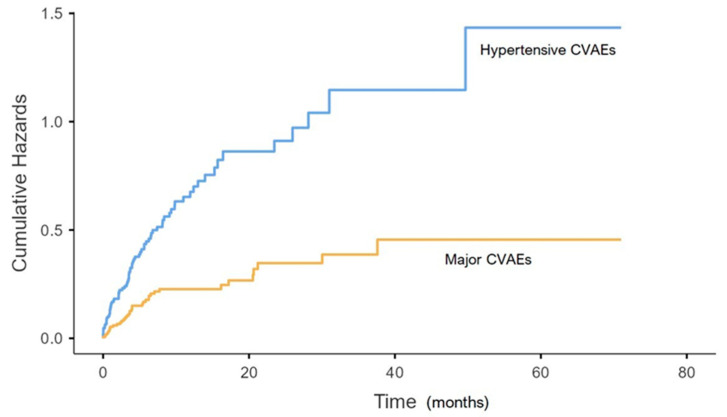
Kaplan–Meier curve of incidence of hypertension-related and major CVAEs. Arterial hypertension-related cardiovascular adverse events (CVAEs) are indicated in blue and major CVAEs in yellow. Patients with both hypertensive and major CVAEs are represented (with the proper onset time) in both curves.

**Figure 2 cancers-15-01149-f002:**
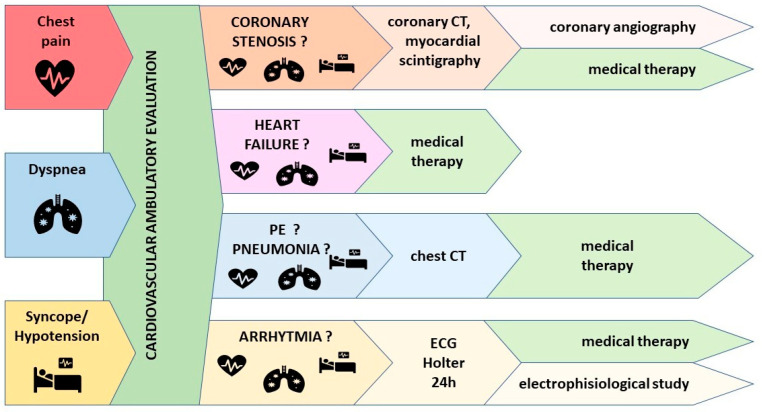
Management protocol of cardiovascular adverse events during CFZ therapy. CT: computed tomography, PE: pulmonary embolism.

**Table 1 cancers-15-01149-t001:** General and clinical characteristics of the population at baseline.

General Characteristics	Population n = 194
Age, years	67.1 ± 8.4
Male sex, n (%)	110 (56.7)
Weight, kg	73.4 ± 14.2
Height, cm	163.1 ± 10.4
BSA, m^2^	1.8 ± 0.2
BMI, kg/m^2^	27.5 ± 4.4
Cardiovascular risk factors, n (%)	
Tobacco use (past/current)	98 (50.5)
Arterial hypertension (history)	100 (51.5)
Obesity (BMI > 30 kg/m^2^)	63 (32.5)
Diabetes mellitus	21 (10.8)
Chronic kidney disease	29 (14.9)
Dyslipidemia	27 (13.9)
Previous cardiovascular events, n (%)	
Previous AF	8 (4.1)
Previous stroke	4 (2.1)
Previous coronary artery disease	6 (3.1)
Anti-hypertensive drugs *, n (%)	95 (48.9)
Beta-blockers	40 (20.6)
ACE-inhibitors/angiotensin receptor blockers	65 (33.5)
Thiazide diuretics/Loop diuretics	36 (18.6)
Mineralcorticoid receptor antagonists	6 (3.1)
Calcium channel blockers	31 (16)
Oncological history	
MM duration, years	4.1 [1.6–6.9]
Relapsed/Refractory MM, n (%)	175 (90.2)
Newly diagnosed MM, n (%)	19 (9.8)
Previous therapy *	
Previous lines of therapy, n	1 [1; 3]
Anthracyclines, n (%)	45 (23.2)
Alkylating agents, n (%)	126 (64.9)
Immunomodulating agents, n (%)	131 (67.5)
Bortezomib, n (%)	136 (70.1)
Auto-transplantation, n (%)	126 (64.9)

* Patients were mostly treated with multiple and combined therapies; hence, tot % amounts to >100. BSA: body surface area, BMI: body mass index, MM: multiple myeloma, AF: atrial fibrillation.

**Table 2 cancers-15-01149-t002:** Baseline hemodynamic and echocardiographic parameters.

Baseline Parameters	Global Population n = 181	Very-High-Risk Population n = 13	*p* Value
Office BP values			
SBP, mmHg	128.8 ± 17.6	139.4 ± 17.3	**0.04**
DBP, mmHg	76.5 ± 11.4	79.4 ± 11.9	0.38
ABPM *			
Daytime SBP, mmHg	125.5 ± 13.2	126.5 ± 15.1	0.82
Daytime DBP, mmHg	74.7 ± 9.2	71.4 ± 8.9	0.25
24 h SBP, mmHg	121.2 ± 12.5	123.2 ± 16	0.64
24 h DBP, mmHg	71.3 ± 8.2	68.9 ± 9.3	0.36
BP variation, mmHg	9.3 ± 3.5	9.6 ± 3.1	0.83
Transthoracic echocardiography			
Left ventricular mass, g/m^2^	87 ± 19.8	123.6 ± 40.4	**<0.001**
LVEF, %	62.5 ± 6.8	55.1 ± 9.3	**<0.001**
GLS †, %	−21.6 ± 2.5	−18.2 ± 2.8	**<0.001**
Arterial stiffness			
PWV ‡, m/s	8.1 ± 1.9	9.9 ± 1.6	**0.006**
CVAEs risk score ^×^	42.2 [32.7; 58.9]	60.1 [51.7; 75.3]	**0.009**

Mean values estimated in * 165 patients; † 164 patients; ‡ 170 patients; ^×^ 124 patients. Statistically significant *p* Value are in bold. BP: Blood pressure, SBP: systolic blood pressure, DBP: diastolic blood pressure, ABPM: ambulatory blood pressure monitoring, LVEF: left ventricle ejection fraction, GLS: global longitudinal strain, PWV: pulse wave velocity.

**Table 3 cancers-15-01149-t003:** Incidence of cardiovascular adverse events (CVAEs) during Carfilzomib therapy.

Cardiovascular Adverse Events *	Population, n = 178
Total CVAEs (%)	95 (48.7)
Events related to arterial hypertension (%)	82 (46)
New onset or worsening of arterial hypertension (%)	69 (38.8)
Arterial hypertension before Carfilzomib infusion: (%)	42 (23.6)
− With subsequent administration (%) − Without subsequent administration (%)	33 (18.5) 19 (10.7)
Arterial hypertension after Carfilzomib infusion (%)	23 (12.9)
Uncontrolled arterial hypertension (>180/100) with symptoms (%)	6 (3.4)
Hypertensive emergency (%)	0 (0)
Major cardiovascular events (%)	37 (20.8)
Dyspnea (%)	7 (3.9)
Arrhythmia (%)	12 (6.7)
Severe hypotension (%)	6 (3.4)
Heart failure (%)	11 (6.2)
Typical chest pain (%)	8 (4.5)
STEMI (%)	1 (0.6)
NSTEMI (%)	7 (3.9)
Syncope (%)	1 (0.6)
Sudden cardiac death (%)	1 (0.6)
Both major and hypertensive events (%)	23 (12.9)

* Defined according to Common Terminology Criteria for Adverse Events (CTCAE) 5.0 [30].

## Data Availability

The data presented in this study are available in this paper.

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
