# Peer review of "Patients with Very High Risk of Cardiovascular Adverse Events during Carfilzomib Therapy: Prevention and Management of Events in a Single Center Experience"

_cancers, 2023, doi:10.3390/cancers15041149_

Round 1

Reviewer 1 Report

Mingrone and coworkers in their current manuscript present an analysis of carfilzomib-associated adverse effects and potential mitigation strategies in a substantial series of patients with multiple myeloma.

Their topic is of high relevance and there is some merit with the paper. Some important questions arise from its current version that need to be addressed.

Major comments:

i.)              The main limitation of this study is the lack of a control group, i.e., non-myeloma subjects of a comparable age group with similar cardiovascular risk factors. If interpreting the results of the six patients undergoing coronary angiography, one would guess they would have become symptomatic w/o Cfz anyway. In light of this, a single-arm study does not contribute enough evidence to support a generally demanding strategy on “all comers” prior to Cfz.

ii.)            Any discussion/interpretation of the competing risk of myeloma deterioration is missing. Patients that are candidates for Cfz always have an incurable disease. If not done in a multivariate analysis, this aspect at least deserves a spotlight in the discussion.

iii.)           The authors describe one fatality in their series attributable to “cardiac arrest”. How can a different cause of death be ruled out? There are many alternative reasons for a sudden death. Pls. clarify

iv.)           It is necessary (section 3.2.2., for instance) to give the overall mortality during Cfz treatment in this series. If the overall mortality is only 0.6%, this is excellent, lower than expected from Cfz clinical studies and should be discussed.

Minor comments:

v.) Introduction, l. 60: “….whose 30.9%..”: pls. rephrase.

vi.) Introduction, l. 64: Probably, the mode of action of the PI is irreversible, not the toxicity in most instances?!? Furthermore cardiovascular risk in one of the aging general population, not just an MM-related comorbidity?!?

vii.) Section 3.2.2., l. 260-262: what do the authors mean with “… myocardial enzymes were dosed”? Pls. rephrase.

Reviewer 2 Report

_Reference #28 does not exist.

_Better stratification of patients risk groups by incorporating the same imaging techniques and cardiac and lipid profile laboratory tests.

Round 2

Reviewer 1 Report

The authors have very thoroughly addressed all concerns raised during the first round of reviews. There are no additional comments.